# PROXIMAL CURRICULUM WITH TASK CORRELATIONS FOR DEEP REINFORCEMENT LEARNING

## ABSTRACT

Curriculum design for reinforcement learning (RL) can speed up an agent's learning process and help it learn to perform well on complex tasks. However, existing techniques typically require domain-specific hyperparameter tuning, involve expensive optimization procedures for task selection, or are suitable only for specific learning objectives. In this work, we consider curriculum design in contextual multi-task settings where the agent's final performance is measured w.r.t. a *target distribution over complex tasks*. We base our curriculum design on the Zone of Proximal Development concept, which has proven to be effective in accelerating the learning process of RL agents for *uniform distribution over all tasks*. We propose a novel curriculum, ProxCoRL, that effectively balances the need for selecting tasks that are not too difficult for the agent while progressing the agent's learning toward the target distribution via leveraging task correlations. We theoretically justify the task selection strategy of ProxCoRL by analyzing a simple learning setting with REINFORCE learner model. Our experimental results across various domains with challenging target task distributions affirm the effectiveness of our curriculum strategy over state-of-the-art baselines in accelerating the training process of deep RL agents.

## 1 INTRODUCTION

Deep reinforcement learning (RL) has shown remarkable success in various fields such as games, continuous control, and robotics, as evidenced by recent advances in the field (Mnih et al., 2015; Lillicrap et al., 2015; Silver et al., 2017; Levine et al., 2016). However, despite these successes, the broader application of RL in real-world domains is often very limited. Specifically, training RL agents in complex environments, such as contextual multi-task settings and goal-based tasks with sparse rewards, still presents significant challenges (Kirk et al., 2021; Andrychowicz et al., 2017; Florensa et al., 2017; Riedmiller et al., 2018).

Curriculum learning has been extensively studied in the context of supervised learning (Weinshall et al., 2018; Zhou & Bilmes, 2018; Elman, 1993; Bengio et al., 2009). Recent research has explored the benefits of using curriculum learning in sequential decision making settings, such as reinforcement learning and imitation learning (Florensa et al., 2017; Riedmiller et al., 2018; Wöhlke et al., 2020; Florensa et al., 2018; Racanière et al., 2020; Klink et al., 2020a;b; Eimer et al., 2021; Kamalaruban et al., 2019; Yengera et al., 2021). The objective of curriculum design in RL is to speed up an agent's learning process and enable it to perform well on complex tasks by exposing it to a personalized sequence of tasks (Narvekar et al., 2020; Portelas et al., 2021; Weng, 2020). To achieve this objective, several works have proposed different curriculum strategies based on different design principles, such as the Zone of Proximal Development (ZPD) (Vygotsky & Cole, 1978; Chaiklin, 2003) and Self-Paced Learning (SPL) (Kumar et al., 2010; Jiang et al., 2015). However, existing techniques typically require domain-specific hyperparameter tuning, involve expensive optimization procedures for task selection, or are suitable only for specific learning objectives, such as uniform performance objectives.

In this work, we investigate curriculum design in contextual multi-task settings with varying degrees of task similarity, where the agent's final performance is measured w.r.t. a *target distribution over complex tasks*. We base our curriculum design on the Zone of Proximal Development concept, which has proven to be effective in accelerating the learning process of RL agents for *uniform*

*distribution over all tasks* (Florensa et al., 2017; Wöhlke et al., 2020; Florensa et al., 2018; Tzannetos et al., 2023). We propose a novel curriculum strategy, PROXCORL, that effectively balances the need for selecting tasks that are neither too hard nor too easy for the agent (according to the ZPD concept) while still progressing its learning toward the target distribution via leveraging task correlations. We have mathematically derived our curriculum strategy by analyzing a specific learning setting. The strengths of our curriculum strategy include its broad applicability to many domains with minimal hyperparameter tuning, computational and sample efficiency, easy integration with deep RL algorithms, and applicability to any target distribution over tasks, not just uniform distribution. Our main results and contributions are as follows:

I. We propose a curriculum strategy, PROXCORL, that effectively trades off the suitable task difficulty level for the agent and the progression towards the target tasks (Section 3).

II. We mathematically derive PROXCORL for the single target task setting with a discrete pool of tasks by analyzing the effect of picking a task on the agent's learning progress in a specific learning scenario (Section 3.1).

III. We propose an extension of PROXCORL that can be applied to a wide range of task spaces and target distributions. This extension can be seamlessly integrated with deep RL frameworks, making it easy to use and apply in various scenarios (Section 3.2).

IV. We empirically demonstrate that the curricula generated with PROXCORL significantly improve the training process of deep RL agents in various environments, matching or outperforming existing state-of-the-art baselines (Section 4).

## 1.1 RELATED WORK

**Curriculum strategies based on Self-Paced Learning (SPL).** In the realm of supervised learning, curriculum strategies leveraging the SPL concept attempt to strike a balance between exposing the learner to all available training examples and selecting examples in which it currently performs well (Kumar et al., 2010; Jiang et al., 2015). In the context of RL, the SPL concept has been adapted by researchers in SPDL (Klink et al., 2020a;b; 2021), SPACE (Eimer et al., 2021), and CURROT (Klink et al., 2022) by controlling the intermediate task distribution with respect to the learner's current training progress. While both SPDL and CURROT involve a setting where the learner's performance is measured w.r.t. a target distribution over the task space (similar to our objective), SPACE operates in a setting where the learner's performance is measured w.r.t. a uniform distribution over the task space. SPDL and CURROT serve as state-of-the-art baselines in our experimental evaluation. The task selection mechanism varies across these methods. SPDL and CURROT operate by solving an optimization problem at each step to select the most relevant task (Klink et al., 2021; 2022). On the other hand, SPACE relies on ranking tasks based on the magnitude of differences in current/previous critic values to choose the task for the next step (Eimer et al., 2021). Furthermore, the work of CURROT Klink et al. (2022) showcase issues about using KL divergence to measure the similarity between task distributions as used in SPDL – instead, they introduce an alternative approach by posing the curriculum design as a constrained optimal transport problem between task distributions. We provide more detailed information on the hyperparameters used in these methods in the appendix.

**Curriculum strategies based on Unsupervised Environment Design (UED).** The UED problem setting involves automatically designing a distribution of environments that adapts to the learning agent (Dennis et al., 2020). UED represents a self-supervised RL paradigm in which an environment generator evolves alongside a student policy to develop an adaptive curriculum learning approach. This approach can be utilized to create increasingly complex environments for training a policy, leading to the emergence of Unsupervised Curriculum Design. PAIRED (Dennis et al., 2020) is an adversarial training technique that solves the problem of the adversary generating unsolvable environments by introducing an antagonist who works with the environment-generating adversary to design environments in which the protagonist receives a low reward. Furthermore, the connections between UED and another related method called PLR (Jiang et al., 2021b) have been explored in (Jiang et al., 2021a; Parker-Holder et al., 2022), resulting in demonstrated improvements over PAIRED. PLR, originally designed for procedural content generation based environments, samples tasks/levels by prioritizing those with higher estimated learning potential when revisited in the future. TD errors are used to estimate a task's future learning potential. Unlike (Jiang et al., 2021a;

Parker-Holder et al., 2022), PLR does not assume control over the environment generation process, requiring only a black box generation process that returns a task given an identifier.

**Curriculum strategies based on ZPD concept.** Effective teaching provides tasks of moderate difficulty (neither too hard nor too easy) for the learner, as formalized by the Zone of Proximal Development (ZPD) concept (Vygotsky & Cole, 1978; Chaiklin, 2003; Oudeyer et al., 2007; Baranes & Oudeyer, 2013; Zou et al., 2019). In the context of RL, several curriculum strategies are based on the ZPD concept, such as selecting the next task randomly from a set of tasks with success rates within a specific range Florensa et al. (2017; 2018). However, the threshold values for success rates require tuning based on the learner's progress and domain. A unified framework for performance-based starting state curricula in RL is proposed by Wöhlke et al. (2020), while Tzannetos et al. (2023) propose a broadly applicable ZPD-based curriculum strategy with minimal hyperparameter tuning and theoretical justifications. Nonetheless, these techniques are generally suitable only for settings where the learner's performance is evaluated using a uniform distribution over all tasks.

**Other automatic curriculum strategies.** Various automatic curriculum generation approaches exist, including: (i) formulating the curriculum design problem as a meta-level Markov Decision Process (Narvekar et al., 2017; Narvekar & Stone, 2019); (ii) learning to generate training tasks similar to a teacher (Dendorfer et al., 2020; Such et al., 2020; Matiisen et al., 2019; Turchetta et al., 2020); (iii) using self-play for curriculum generation (Sukhbaatar et al., 2018); (iv) leveraging disagreement between different agents trained on the same tasks (Zhang et al., 2020); and (v) selecting starting states based on a single demonstration (Salimans & Chen, 2018; Resnick et al., 2018). Interested readers can refer to recent surveys on RL curriculum design (Narvekar et al., 2020; Portelas et al., 2021; Weng, 2020).

**Curriculum strategies based on domain knowledge.** In supervised learning, early works involve ordering examples by increasing difficulty (Elman, 1993; Bengio et al., 2009; Schmidhuber, 2013; Zaremba & Sutskever, 2014), which has been adapted in hand-crafted RL curriculum approaches (Asada et al., 1996; Wu & Tian, 2016). Recent works on imitation learning have also utilized iterative machine teaching framework to design greedy curriculum strategies (Kamalaruban et al., 2019; Yengera et al., 2021; Liu et al., 2017; Yang et al., 2018; Zhu et al., 2018). However, all these approaches require domain-specific expert knowledge for designing difficulty measures.

## 2 FORMAL SETUP

In this section, we formalize our problem setting based on prior work on teacher-student curriculum learning (Matiisen et al., 2019).

**Multi-task RL.** We consider a multi-task RL setting with a task/context space $\mathcal{C}$, in which each task $c \in \mathcal{C}$ is associated with a learning environment modeled as a contextual Markov Decision Process (MDP), denoted by $\mathcal{M}_c := \big(\mathcal{S}, \mathcal{A}, \gamma, \mathcal{T}_c, R_c, P_c^0\big)$ (Hallak et al., 2015; Modi et al., 2018). The state space $\mathcal{S}$ and action space $\mathcal{A}$ are shared by all tasks in $\mathcal{C}$, as well as the discount factor $\gamma$. Each contextual MDP includes a contextual transition dynamics $\mathcal{T}_c : \mathcal{S} \times \mathcal{S} \times \mathcal{A} \rightarrow [0, 1]$, a contextual reward function $R_c : \mathcal{S} \times \mathcal{A} \rightarrow [-R_{\max}, R_{\max}]$, where $R_{\max} > 0$, and a contextual initial state distribution $P_c^0 : \mathcal{S} \rightarrow [0, 1]$. We denote the space of environments by $\mathcal{M} = \{\mathcal{M}_c : c \in \mathcal{C}\}$.

**RL agent and training process.** We consider an RL agent acting in any environment $\mathcal{M}_c \in \mathcal{M}$ via a contextual policy $\pi : \mathcal{S} \times \mathcal{C} \times \mathcal{A} \rightarrow [0, 1]$ that is a contextual mapping from a state to a probability distribution over actions. Given a task $c \in \mathcal{C}$, the agent attempts the task via a trajectory rollout obtained by executing its policy $\pi$ in the MDP $\mathcal{M}_c$. The trajectory rollout is denoted as $\xi = \big\{(s^{(\tau)}, a^{(\tau)})\big\}_{\tau=0,1,\ldots}$ with $s^{(0)} \sim P_c^0$. The agent's performance on task $c$ is measured by the value function $V^\pi(c) := \mathbb{E}\big[\sum_{\tau=0}^\infty \gamma^\tau \cdot R_c(s^{(\tau)}, a^{(\tau)}) \big| \pi, \mathcal{M}_c\big]$. The agent training corresponds to finding a policy that performs well w.r.t. a target distribution $\mu$ over $\mathcal{C}$, i.e., $\max_\pi V_\mu^\pi$ where $V_\mu^\pi := \mathbb{E}_{c \sim \mu}[V^\pi(c)]$. The training process of the agent involves an interaction between two components: a student component that is responsible for policy updates and a teacher component that is responsible for task selection. The interaction happens in discrete steps indexed by $t = 1, 2, \ldots$, and is formally described in Algorithm 1. Let $\pi_{\text{end}}$ denote the agent's final policy at the end of teacher-student interaction. The *training objective* is to ensure that the performance of the policy $\pi_{\text{end}}$ is $\epsilon$-near-optimal, i.e., $(\max_\pi V_\mu^\pi - V_\mu^{\pi_{\text{end}}}) \leq \epsilon$.

---

**Algorithm 1** RL Agent Training as Interaction between Teacher-Student Components

---

1: **Input:** RL agent's initial policy $\pi_1$
2: **for** $t = 1, 2, \ldots$ **do**
3:      Teacher component picks a task $c_t \in \mathcal{C}$.
4:      Student component attempts the task via a trajectory rollout $\xi_t$ using the policy $\pi_t$ in $\mathcal{M}_{c_t}$.
5:      Student component updates the policy to $\pi_{t+1}$ using the rollout $\xi_t$.
6: **Output:** RL agent's final policy $\pi_{\text{end}} \leftarrow \pi_{t+1}$.

---

**Student component.** We consider a parametric representation for the RL agent, whose current knowledge is parameterized by $\theta \in \Theta \subseteq \mathbb{R}^d$, and each parameter $\theta$ is mapped to a policy $\pi_\theta : \mathcal{S} \times \mathcal{C} \times \mathcal{A} \rightarrow [0, 1]$. At step $t$, the student component updates the knowledge parameter based on the following quantities: the current knowledge parameter $\theta_t$, the task $c_t$ picked by the teacher component, and the rollout $\xi_t = \left\{ (s_t^{(\tau)}, a_t^{(\tau)}) \right\}_\tau$. Then, the updated knowledge parameter $\theta_{t+1}$ is mapped to the agent's policy given by $\pi_{t+1} := \pi_{\theta_{t+1}}$. As a concrete example, for the REINFORCE agent (Sutton et al., 1999), the knowledge parameter is updated as follows: $\theta_{t+1} \leftarrow \theta_t + \eta_t \cdot \sum_{\tau=0}^\infty G_t^{(\tau)} \cdot g_t^{(\tau)}$, where $\eta_t$ is the learning rate, $G_t^{(\tau)} = \sum_{\tau'=\tau}^\infty \gamma^{\tau'-\tau} \cdot R_{c_t}(s_t^{(\tau')}, a_t^{(\tau')})$, and $g_t^{(\tau)} = \left[ \nabla_\theta \log \pi_\theta(a_t^{(\tau)} | s_t^{(\tau)}, c_t) \right]_{\theta=\theta_t}$.

**Teacher component.** At time step $t$, the teacher component selects a task $c_t$ for the student component to attempt via a trajectory rollout, as shown in line 3 in Algorithm 1. The sequence of tasks, also known as the curriculum, that is chosen by the teacher component has a significant impact on the performance improvement of the policy $\pi_t$. The primary objective of this work is to develop a teacher component to achieve the training objective in a computationally efficient and sample-efficient manner.

## 3 OUR CURRICULUM STRATEGY PROXCORL

In Section 3.1, we mathematically derive a curriculum strategy for the single target task setting with a discrete pool of tasks by analyzing a specific learning scenario. Then, in Section 3.2, we present our final curriculum strategy that is applicable in general learning settings.

### 3.1 CURRICULUM STRATEGY FOR SINGLE TARGET TASK SETTINGS

In this section, we present our curriculum strategy for a setting where the task space $\mathcal{C}$ is a discrete set and the target distribution $\mu$ is a delta distribution concentrated on a single target task $c_{\text{targ}}$. To design our curriculum strategy, we investigate the effect of selecting a task $c_t$ at time step $t$ on the agent's performance $V_\mu^{\pi_{\theta_t}}$ and its convergence towards the target performance $V_\mu^* := \max_\pi V_\mu^\pi$. Therefore, we define the training objective improvement at time step $t$ and analyze this metric for a specific learning scenario.

**Expected improvement in the training objective.** At time step $t$, given the current knowledge parameter $\theta_t$, the task $c_t$ picked by the teacher component, and the student component's rollout $\xi_t$, we define the improvement in the training objective as follows:

$$\Delta_t(\theta_{t+1} | \theta_t, c_t, \xi_t) := (V_\mu^* - V_\mu^{\pi_{\theta_t}}) - (V_\mu^* - V_\mu^{\pi_{\theta_{t+1}}}).$$

Additionally, we define the expected improvement in the training objective at step $t$ due to picking the task $c_t$ as follows (Weinshall et al., 2018; Kamalaruban et al., 2019; Yengera et al., 2021; Graves et al., 2017):

$$I_t(c_t) := \mathbb{E}_{\xi_t | c_t} [\Delta_t(\theta_{t+1} | \theta_t, c_t, \xi_t)] .$$

Based on the above measure, a natural greedy curriculum strategy for selecting the next task $c_t$ is given by: $c_t \leftarrow \arg\max_{c \in \mathcal{C}} I_t(c)$. We aim to approximate such a curriculum strategy without computing the updated policy $\pi_{\theta_{t+1}}$. To this end, we analyze the function $I_t(\cdot)$ for REINFORCE learner model under a specific learning setting. This analysis enables us to develop an intuitive curriculum strategy by effectively combining three fundamental factors: (i) the learning potential inherent in the source task, (ii) the transfer potential between the source and target tasks, i.e., their

similarity, and (iii) the potential for performance improvement in the target task. Looking ahead, it is indeed captivating to extend our investigations to more complex learning settings, where we can explore the potential for devising more sophisticated curriculum strategies.

**Intuitive form of $I_t(\cdot)$.** We define $g_t : \mathcal{C} \to \mathbb{R}^d$ as $g_t(c) := [\nabla_\theta V^{\pi_\theta}(c)]_{\theta=\theta_t}$, and $\psi_t : \mathcal{C} \to \mathbb{R}^d$ as $\psi_t(c) := \frac{g_t(c)}{\|g_t(c)\|}$. By applying the first-order Taylor approximation of $V^{\pi_{\theta_{t+1}}}(c_{\text{targ}})$ at $\theta_t$, we approximate the improvement in the training objective as follows:

$$\Delta_t(\theta_{t+1} | \theta_t, c_t, \xi_t) = V^{\pi_{\theta_{t+1}}}(c_{\text{targ}}) - V^{\pi_{\theta_t}}(c_{\text{targ}}) \approx \langle \theta_{t+1} - \theta_t, g_t(c_{\text{targ}}) \rangle.$$

The knowledge parameter update for the REINFORCE agent can be written as: $\theta_{t+1} \leftarrow \theta_t + \eta_t \cdot \widehat{g_t(c_t)}$, where $\mathbb{E}_{\xi_t | c_t}\left[\widehat{g_t(c_t)}\right] = g_t(c_t)$. Then, for the REINFORCE agent, we approximate the expected improvement in the training objective as follows:

$$I_t(c_t) \approx \langle \mathbb{E}_{\xi_t | c_t}[\theta_{t+1} - \theta_t], g_t(c_{\text{targ}}) \rangle = \eta_t \cdot \|g_t(c_t)\| \cdot \|g_t(c_{\text{targ}})\| \cdot \langle \psi_t(c_t), \psi_t(c_{\text{targ}}) \rangle.$$

In the above, the term $\|g_t(c_t)\|$ corresponds to the learning potential inherent in the source task, the term $\|g_t(c_{\text{targ}})\|$ corresponds to the learning potential inherent in the target task, and the term $\langle \psi_t(c_t), \psi_t(c_{\text{targ}}) \rangle$ corresponds to the transfer potential between the source and target tasks. In the subsequent discussion, we analyze the function $I_t(\cdot)$ under a contextual bandit setting.

**Contextual bandit setting.** We consider the REINFORCE learner model with the following policy parameterization: given a feature mapping $\phi : \mathcal{S} \times \mathcal{C} \times \mathcal{A} \to \mathbb{R}^d$, for any $\theta \in \mathbb{R}^d$, we parameterize the policy as $\pi_\theta(a|s,c) = \frac{\exp(\langle \theta, \phi(s,c,a) \rangle)}{\sum_{a'} \exp(\langle \theta, \phi(s,c,a') \rangle)}, \forall s \in \mathcal{S}, c \in \mathcal{C}, a \in \mathcal{A}$. In the following, we consider a specific problem instance of contextual MDP setting. Let $\mathcal{M}_c$ be a contextual MDP with a singleton state space $\mathcal{S} = \{s\}$, and an action space $\mathcal{A} = \{a_1, a_2\}$. Any action $a \in \mathcal{A}$ taken from the initial state $s \in \mathcal{S}$ always leads to a terminal state. Let $r : \mathcal{C} \to [0, 1]$ be a mapping from task/context space $\mathcal{C}$ to the interval $[0, 1]$. For any context $c \in \mathcal{C}$, we denote the optimal and non-optimal actions for that context as $a_c^{\text{opt}}$ and $a_c^{\text{non}}$, respectively. The contextual reward function is defined as follows: $R_c(s, a_c^{\text{opt}}) = 1$, and $R_c(s, a_c^{\text{non}}) = 0$, for all $c \in \mathcal{C}$. Further, we define $\psi : \mathcal{C} \to \mathbb{R}^d$ as $\psi(c) := (\phi(s, c, a_c^{\text{opt}}) - \phi(s, c, a_c^{\text{non}}))$. Subsequently, for the REINFORCE agent operating under the above setting, the following theorem quantifies the expected improvement in the training objective at time step $t$:

**Theorem 1.** *For the* REINFORCE *agent with softmax policy parameterization under the contextual bandit setting described above, we have:*

$$I_t(c) \approx \eta_t \cdot \frac{V^{\pi_{\theta_t}}(c)}{V^*(c)} \cdot \left(V^*(c) - V^{\pi_{\theta_t}}(c)\right) \cdot \frac{V^{\pi_{\theta_t}}(c_{\text{targ}})}{V^*(c_{\text{targ}})} \cdot \left(V^*(c_{\text{targ}}) - V^{\pi_{\theta_t}}(c_{\text{targ}})\right) \cdot \langle \psi(c), \psi(c_{\text{targ}}) \rangle,$$

*where $V^*(c) = \max_\pi V^\pi(c)$, and $\eta_t$ is the learning of the* REINFORCE *agent.*

A detailed proof of Theorem 1 can be found in Appendix C.1. As a more general result, we conducted an analysis of the gradient function $g_t(\cdot)$ within the context of a tree-structured contextual MDP setting, as described in Appendix C.2. This analysis establishes a connection between $\|g_t(c)\|_1$ and the term $\frac{V^{\pi_{\theta_t}}(c)}{V^*(c)} \cdot \left(V^*(c) - V^{\pi_{\theta_t}}(c)\right)$.

**Curriculum strategy.** Inspired by the above analysis, we propose the following curriculum strategy:

$$c_t \leftarrow \underset{c \in \mathcal{C}}{\arg\max} \underbrace{\frac{V^{\pi_{\theta_t}}(c)}{V^*(c)}}_{①} \cdot \underbrace{\left(V^*(c) - V^{\pi_{\theta_t}}(c)\right)}_{②} \cdot \underbrace{\left(V^*(c_{\text{targ}}) - V^{\pi_{\theta_t}}(c_{\text{targ}})\right)}_{③} \cdot \underbrace{\langle \psi(c), \psi(c_{\text{targ}}) \rangle}_{④}, \quad (1)$$

where $V^*(c) = \max_\pi V^\pi(c)$ and $\psi : \mathcal{C} \to \mathbb{R}^d$ is a context representation mapping. Given that the term $\frac{V^{\pi_{\theta_t}}(c_{\text{targ}})}{V^*(c_{\text{targ}})}$ tends to have a significantly low value, we omit its inclusion in the above proposal for the sake of numerical stability. At time step $t$, the teacher component picks a task $c_t$ according to the above equation. The curriculum strategy involves the following quantities: ① the agent's relative performance on the task $c$, ② the expected regret of the agent on the task $c$, ③ the expected regret of the agent on the target task $c_{\text{targ}}$, and ④ the correlation between the tasks $c$ and $c_{\text{targ}}$. The product of the terms ① and ② enforces picking tasks that are neither too hard nor too easy for the current policy (corresponding to the ZPD principle). The product of the terms ③ and ④ enforces picking tasks that are highly correlated with the target task. The curriculum strategy effectively balances these two objectives.

|  | SGR | POINTMASS-s:2G | POINTMASS-s:1T | BIPEDALWALKER |
|---|---|---|---|---|
| Reward | binary | binary | binary | dense |
| Context | $\mathbb{R}^3$ | $\mathbb{R}^3$ | $\mathbb{R}^3$ | $\mathbb{R}^2$ |
| State | $\mathbb{R}^4$ | $\mathbb{R}^4$ | $\mathbb{R}^4$ | $\mathbb{R}^{24}$ |
| Action | $\mathbb{R}^2$ | $\mathbb{R}^2$ | $\mathbb{R}^2$ | $\mathbb{R}^4$ |
| Target Dist. | $\mathbb{R}^2$ Plane | Double-Mode Gaussian | Single Task | Uniform with trivial tasks |

(a) Complexity of environments

(b) Illustration of the environments

Figure 1: **(a)** provides a comprehensive overview of the complexity of the environments based on the reward signals, context space, state space, action space, and target distribution. **(b)** showcases the environments by providing an illustrative visualization of each environment (from left to right): SGR, POINTMASS-s and BIPEDALWALKER.

## 3.2 CURRICULUM STRATEGY FOR GENERAL SETTINGS

In this section, we extend the curriculum strategy in Eq. (1) to practical settings of interest, i.e., a general task space $\mathcal{C}$, a general target distribution $\mu$, and $V^*(c)$ values being unknown. We begin by constructing two large discrete sets, $\widehat{\mathcal{C}}_{\text{unif}}$ and $\widehat{\mathcal{C}}_{\text{targ}}$, which are subsets of the original task space $\mathcal{C}$. $\widehat{\mathcal{C}}_{\text{unif}}$ is obtained by sampling contexts from $\mathcal{C}$ according to uniform distribution, while $\widehat{\mathcal{C}}_{\text{targ}}$ is obtained by sampling contexts from $\mathcal{C}$ according to the target distribution $\mu$. For the general setting, we consider the following curriculum strategy:

$$(c_{\text{targ}}^t, c_t) \leftarrow \underset{(c_{\text{targ}}, c) \in \widehat{\mathcal{C}}_{\text{targ}} \times \widehat{\mathcal{C}}_{\text{unif}}}{\arg\max} \frac{V^{\pi_{\theta_t}}(c)}{V^*(c)} \cdot \left(V^*(c) - V^{\pi_{\theta_t}}(c)\right) \cdot \left(V^*(c_{\text{targ}}) - V^{\pi_{\theta_t}}(c_{\text{targ}})\right) \cdot \langle \psi(c), \psi(c_{\text{targ}}) \rangle. \tag{2}$$

Next, we replace $V^*(\cdot)$ with $V_{\max}$, i.e., the maximum possible value that can be achieved for any task in the task space – this value can typically be obtained for a given domain. Further, when training deep RL agents, allowing some stochasticity in task selection is useful. In particular, the $\arg\max$ selection in Eq. (2) can be problematic in the presence of any approximation errors while computing $V^{\pi_{\theta_t}}(\cdot)$ values. To make the selection more robust, we replace $\arg\max$ selection in Eq. (2) with softmax selection and sample $(c_{\text{targ}}^t, c_t)$ from the distribution given below:

$$\mathbb{P}\left[(c_{\text{targ}}^t, c_t) = (c_{\text{targ}}, c)\right] \propto \exp\left(\beta \cdot \frac{V^t(c)}{V_{\max}} \cdot \left(V_{\max} - V^t(c)\right) \cdot \left(V_{\max} - V^t(c_{\text{targ}})\right) \cdot \langle \psi(c), \psi(c_{\text{targ}}) \rangle\right), \tag{3}$$

where $\beta$ is a hyperparameter and $V^t(\cdot)$ values are obtained from the critic network of the RL agent to estimate $V^{\pi_{\theta_t}}(\cdot)$. Finally, the teacher component samples $(c_{\text{targ}}^t, c_t)$ from the above distribution and provides the task $c_t$ to the student component – we refer to this selection strategy as PROXCORL.

## 4 EXPERIMENTAL EVALUATION

In this section, we validate the effectiveness of our curriculum strategy by conducting experiments in environments selected from the state-of-the-art works of Klink et al. (2022) and Romac et al. (2021). Throughout the experiments, we utilize the PPO method from the Stable-Baselines3 library for policy optimization (Schulman et al., 2017; Raffin et al., 2021).

### 4.1 ENVIRONMENTS

In our evaluation, we examine three distinct environments detailed in the following paragraphs. These environments are selected to showcase the effectiveness of our curriculum strategy in handling target distributions with varying characteristics within the context space $\mathcal{C}$. The first environment, *Sparse Goal Reaching* (SGR), features target distributions with uniform coverage over specific dimensions of the context space and concentrated on one dimension. For the second environment, *Point Mass Sparse* (POINTMASS-s) we consider two settings. In one setting, the target distribution exhibits multiple modalities. In the second setting, the target is concentrated on

a single context $c \in \mathcal{C}$. Lastly, the third environment has a uniform target distribution spanning the entirety of the context space. A summary and illustration of these environments are presented in Figure 1. For additional details about each environment, please refer to Appendix D.1.

***Sparse Goal Reaching* (SGR).** Based on the work of Klink et al. (2022), we consider a sparse-reward, goal-reaching environment in which an agent needs to reach a desired position with high precision. Such environments have previously been studied by Florensa et al. (2018). Within this environment, the contexts, denoted as $c \in \mathcal{C} \subseteq \mathbb{R}^3$, encode both the desired 2D goal position and the acceptable tolerance for reaching that goal. Our primary objective centers around achieving as many goals as possible with high precision, indicated by a low tolerance threshold. In this regard, the target distribution $\mu$ takes the form of a uniform distribution, but it is restricted to a specific 2D region within $\mathcal{C}$ where the tolerance (*C-Tolerance*) for each context is set at a minimal value of $0.05$. Additionally, the presence of walls within the environment renders many of the tasks specified by $\mathcal{C}$ infeasible, necessitating the identification of a feasible task subspace. We generate our training tasks by randomly selecting 9900 contexts from $\mathcal{C}$ using uniform distribution to create $\widehat{\mathcal{C}}_{\text{unif}}$, and by selecting 100 contexts according to the target distribution $\mu$ to form $\widehat{\mathcal{C}}_{\text{targ}}$. For the purpose of evaluation, we employ a separate held-out set sampled from the target distribution $\mu$.

***Point Mass Sparse* (POINTMASS-S).** Based on the work of Klink et al. (2020b), we consider a contextual POINTMASS-S environment where an agent navigates a point mass through a gate of a given size towards a goal in a two-dimensional space. To heighten the challenge, we replace the original dense reward function with a sparse one, a strategy also considered in Tzannetos et al. (2023). Specifically, in the POINTMASS-S environment, the agent operates within a goal-based reward setting where the reward is binary and sparse, i.e., the agent receives a reward of 1 only upon successfully moving the point mass to the goal position. The parameters governing this environment, such as the gate's position, width, and the ground's friction coefficient, are controlled by a contextual variable $c \in \mathcal{C} \subseteq \mathbb{R}^3$. This variable comprises *C-GatePosition*, *C-GateWidth*, and *C-Friction*. Our experimental section explores two distinct POINTMASS-S environment settings. In the first setting, denoted as POINTMASS-S:2G, the target distribution $\mu$ takes the form of a bimodal Gaussian distribution. Here, the means of the contextual variables $\big[$*C-GatePosition*, *C-GateWidth*$\big]$ are set to $\big[-3.9, 0.5\big]$ and $\big[3.9, 0.5\big]$ for the two modes, respectively. In the second setting, POINTMASS-S:1T, the target distribution $\mu$ is concentrated on a single context $c \in \mathcal{C}$. More precisely, the contextual variables $\big[$*C-GatePosition*, *C-GateWidth*, *C-Friction*$\big]$ take on the following values: $\big[0.9, 0.5, 3.5\big]$. To construct our training tasks, we draw 20000 contexts from $\mathcal{C}$ using a uniform distribution, forming $\widehat{\mathcal{C}}_{\text{unif}}$. The set $\widehat{\mathcal{C}}_{\text{targ}}$ is created by sampling 400 contexts from $\mathcal{C}$ according to the target distribution $\mu$. We employ a held-out set sampled from the target distribution $\mu$ for evaluation purposes.

***Bipedal Walker Stump Tracks* (BIPEDALWALKER).** We conduct additional experiments within the TeachMyAgent benchmark for curriculum techniques, as introduced in Romac et al. (2021). In this context, we chose a bipedal agent tasked with walking in the Stump Tracks environment, which is an extension of the environment initially proposed in Portelas et al. (2019). The state space comprises lidar sensors, head position, and joint positions. The action space is continuous, and the goal is to learn a policy that controls the torque of the agent's motors. The walker is rewarded for going forward and penalized for torque usage. An episode lasts 2000 steps and is terminated if the agent reaches the end of the track or if its head collides with the environment (in which case a reward of $-100$ is received). Within this environment, the contextual variables $c \in \mathcal{C} \subseteq \mathbb{R}^2$ control the height (*C-StumpHeight*) and spacing (*C-StumpSpacing*) of stumps placed along the track for each task. Our experimental setup is equivalent to the bipedal walker stump track environment with mostly trivial tasks, as described in Romac et al. (2021). In this setup, *C-StumpHeight* is constrained to the range $\big[-3; 3\big]$, while *C-StumpSpacing* remains within $\big[0; 6\big]$. Notably, the environment enforces the clipping of negative values for *C-StumpHeight*, setting them to $0$. Consequently, half of the tasks have a mean stump height of $0$, introducing a significant proportion of trivial tasks ($50\%$). To address the procedural task generation, we randomly draw 1000 tasks from $\mathcal{C}$ to construct the training task set, denoted as $\widehat{\mathcal{C}}_{\text{unif}}$. Additionally, every four epochs, we resample 1000 tasks and update the training set $\widehat{\mathcal{C}}_{\text{unif}}$. The set $\widehat{\mathcal{C}}_{\text{targ}}$ is obtained by sampling 500 tasks from $\mathcal{C}$ according to the target distribution $\mu$, which is uniform in $\mathcal{C}$.

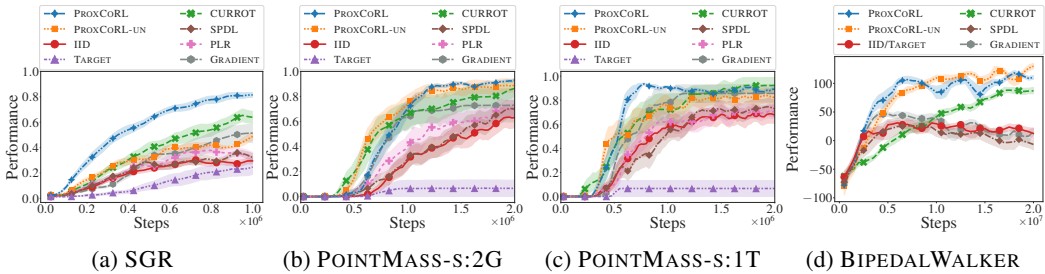

(a) SGR     (b) PointMass-s:2G     (c) PointMass-s:1T     (d) BipedalWalker

Figure 2: Performance comparison of RL agents trained using different curriculum strategies. The performance is measured as the mean return ($\pm 1$ standard error) on the test pool of tasks. The results are averaged over 10 random seeds for SGR, 15 random seeds for PointMass-s:2G, 15 random seeds for PointMass-s:1T and 10 random seeds for BipedalWalker. The plots are smoothed across 2 evaluation snapshots that occur over 25000 training steps.

## 4.2 Curriculum Strategies Evaluated

**Variants of our curriculum strategy.** We consider two curriculum strategies as described next. First, ProxCoRL is based on Eq. (3). Throughout all the experiments, we use the following choice to compute similarity between $\psi(s)$ and $\psi(c_{\text{targ}})$: $\exp(-||c - c_{targ}||_2)$. Second, ProxCoRL-un is a variant of it that does not take into account the target distribution $\mu$ and hence ignores the correlations. Specifically, ProxCoRL-un drops the target task-related terms ③ and ④ derived in Eq.(1), and selects the next task according to the following distribution: $\mathbb{P}\big[c_t = c\big] \propto \exp\big(\beta \cdot \frac{V^t(c)}{V_{\max}} \cdot (V_{\max} - V^t(c))\big)$. We note that this strategy is similar to a ZPD-based curriculum strategy proposed in Tzannetos et al. (2023) for uniform performance objective.

**State-of-the-art baselines.** SPDL (Klink et al., 2020b), CURROT (Klink et al., 2022), PLR (Jiang et al., 2021b), and Gradient (Huang et al., 2022) are state-of-the-art curriculum strategies for contextual RL. We adapt the implementation of an improved version of SPDL, presented in Klink et al. (2021), to work with a discrete pool of contextual tasks. PLR (Jiang et al., 2021b) was originally designed for procedurally generated content settings, but we have adapted its implementation for the contextual RL setting operating on a discrete pool of tasks.

**Prototypical baselines.** We consider two prototypical baselines: IID and Target. The IID strategy samples the next task from $\mathcal{C}$ with a uniform distribution, while the Target strategy samples according to the target distribution $\mu$.

## 4.3 Results

**Convergence behavior.** As illustrated in Figure 2, the RL agents trained using our curriculum strategy, ProxCoRL, perform competitively w.r.t. those trained with state-of-the-art and prototypical baselines. In Figure 2a for SGR, ProxCoRL outperforms all the other techniques by a large margin. ProxCoRL selects tasks that are neither too hard nor too easy for the agent's current policy and are also correlated with the target distribution. CURROT stands out among other strategies due to its ability to gradually choose tasks from the target distribution. Importantly, solely selecting target contexts for training is inadequate, as evidenced by the underperformance of Target compared to all other techniques. The results for PointMass-s:2G are presented in Figure 2b, where we can observe that ProxCoRL, ProxCoRL-un, and CURROT outperform the other strategies. ProxCoRL demonstrates success in handling bimodal target distributions by alternating the selection between the modes of the target distribution. Although it initially has a slower performance than ProxCoRL-un and CURROT, it eventually matches/surpasses their performance. Despite ProxCoRL-un not explicitly considering the target distribution in its formulation, it progressively selects more challenging contexts and effectively encompasses the tasks from the target distribution in this scenario. For PointMass-s:1T, in Figure 2c, we observe that ProxCoRL quickly succeeds in the single target task compared to the other techniques. Although CURROT converges slower, it finally performs similarly to the proposed technique. For BipedalWalker, Figure 2d, where the target distribution is uniform, ProxCoRL-un achieves the best performance. This technique, by definition, considers a uniform performance objective. ProxCoRL is able to handle a uniform target distribution better than CURROT for this setting.

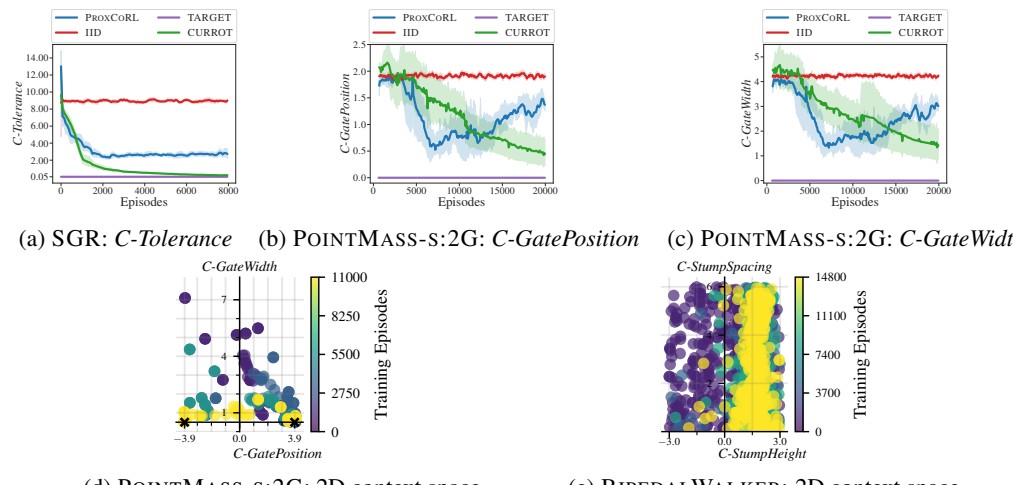

(a) SGR: *C-Tolerance*    (b) POINTMASS-S:2G: *C-GatePosition*    (c) POINTMASS-S:2G: *C-GateWidth*

(d) POINTMASS-S:2G: 2D context space      (e) BIPEDALWALKER: 2D context space

Figure 3: **(a)** presents the average *C-Tolerance* of the selected tasks during different curriculum strategies for SGR. **(b-c)** present the average distance between the selected contexts *C-GatePosition* and *C-GateWidth* and the target distribution for POINTMASS-S:2G. **(d)** presents the two-dimensional context space of POINTMASS-S:2G. The target distribution is depicted as a black **x** and encodes the two gates with *C-GateWidth* = 0.5 at *C-GatePosition* = {−3.9, 3.9}. Each colored dot represents the context/task selected by PROXCORL during training, where brighter colors indicate later training stages. **(e)** presents the two-dimensional context space of BIPEDALWALKER. The target distribution is uniform. Each colored dot represents the context/task selected by PROXCORL during training, where brighter colors indicate later training stages.

**Curriculum plots.** Figure 3a displays the average *C-Tolerance* of tasks selected from the proposed curriculum (PROXCORL), CURROT, IID, and TARGET. Our observation reveals that both PROXCORL and CURROT manage to reduce the average *C-Tolerance* below that of IID, indicating that both techniques gradually prioritize tasks that align with the target distribution. However, it is noteworthy that CURROT continues to decrease the context values to reach the target distribution, while PROXCORL does not necessarily converge to the target distribution. This trend is similarly evident in Figures 3b and 3c, where PROXCORL after succeeding on the target distribution returns to sampling closer to IID. Conversely, CURROT persists in reducing the context values to attain convergence with the target distribution. Figure 3d provides a visual representation of the two-dimensional context space for the POINTMASS-S:2G setting. The curriculum initially starts from larger *C-GateWidth* values and centered *C-GatePosition* values, gradually shifting towards the two modes of the target distribution in the later stages of training. In Figure 3e, we depict the two-dimensional context space for the BIPEDALWALKER setting. Despite the uniformity of the target distribution of contexts, we observe that in the later stages of training, PROXCORL disregards trivial tasks characterized by *C-StumpHeight* values smaller than 0. Instead, it focuses on tasks from the remaining task space.

## 5 CONCLUDING DISCUSSIONS

We proposed a novel curriculum strategy that strikes a balance between selecting tasks that are neither too hard nor too easy for the agent while also progressing the agent's learning toward the target distribution by utilizing task correlation. We mathematically derived our curriculum strategy through an analysis of a specific learning scenario and demonstrated its effectiveness in various environments through empirical evaluations. Here, we discuss a few limitations of our work and outline a plan on how to address them in future work. First, it would be interesting to extend our curriculum strategy to high-dimensional context spaces in sparse reward environments. However, sampling new tasks in such environments poses a significant challenge due to the estimation of the value of all tasks in the discrete sets $\widehat{\mathcal{C}}_{\mathrm{unif}}$ and $\widehat{\mathcal{C}}_{\mathrm{targ}}$. Second, while our curriculum strategy uses a simple distance measure to capture task correlation, it would be worthwhile to investigate the effects of employing different distance metrics over the context space on curriculum design.

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

## A    TABLE OF CONTENTS

In this section, we briefly describe the content provided in the paper's appendices.

- Appendix B provides a discussion of the broader impact of our work and compute resources used.
- Appendix C provides a proof for Theorem 1 and an additional theoretical justification for our curriculum strategy. (Section 3.1)
- Appendix D provides additional details for experimental evaluation. (Section 4)

## B    DISCUSSIONS

**Broader impact.** This work presents a novel curriculum strategy for contextual multi-task settings where the agent's final performance is measured w.r.t. a target distribution over the context space. Given the algorithmic and empirical nature of our work applied to learning agents, we do not foresee any direct negative societal impacts of our work in the present form.

**Compute resources.** The experiments for SGR and POINTMASS-S were conducted on a cluster of machines equipped with Intel Xeon Gold 6134M CPUs running at a frequency of 3.20GHz. The experiments for BIPEDALWALKER were conducted on machines equipped with Tesla V100 GPUs.

## C    JUSTIFICATIONS FOR THE CURRICULUM STRATEGY

In this section, we provide some insights into the proposed curriculum strategy by analyzing two specific learning settings. For both settings, we consider the REINFORCE learner model with the following policy parameterization: given a feature mapping $\phi : \mathcal{S} \times \mathcal{C} \times \mathcal{A} \to \mathbb{R}^d$, for any $\theta \in \mathbb{R}^d$, we parameterize the policy as $\pi_\theta(a|s,c) = \frac{\exp(\langle \theta, \phi(s,c,a) \rangle)}{\sum_{a'} \exp(\langle \theta, \phi(s,c,a') \rangle)}, \forall s \in \mathcal{S}, c \in \mathcal{C}, a \in \mathcal{A}$.

### C.1    CONTEXTUAL BANDIT SETTING

Let $\mathcal{M}_c$ be a contextual MDP with a singleton state space $\mathcal{S} = \{s\}$, and an action space $\mathcal{A} = \{a_1, a_2\}$. Any action $a \in \mathcal{A}$ taken from the initial state $s \in \mathcal{S}$ always leads to a terminal state. Let $r : \mathcal{C} \to [0,1]$ be a mapping from task/context space $\mathcal{C}$ to the interval $[0,1]$. For any context $c \in \mathcal{C}$, we denote the optimal and non-optimal actions for that context as $a_c^{\text{opt}}$ and $a_c^{\text{non}}$, respectively. The contextual reward function is defined as follows: $R_c(s, a_c^{\text{opt}}) = 1$, and $R_c(s, a_c^{\text{non}}) = 0$, for all $c \in \mathcal{C}$. Further, we define $\psi : \mathcal{C} \to \mathbb{R}^d$ as $\psi(c) := (\phi(s,c,a_c^{\text{opt}}) - \phi(s,c,a_c^{\text{non}}))$.

**Theorem 1.** *For the* REINFORCE *agent with softmax policy parameterization under the contextual bandit setting described above, we have:*

$$I_t(c) \approx \eta_t \cdot \frac{V^{\pi_{\theta_t}}(c)}{V^*(c)} \cdot \left( V^*(c) - V^{\pi_{\theta_t}}(c) \right) \cdot \frac{V^{\pi_{\theta_t}}(c_{\text{targ}})}{V^*(c_{\text{targ}})} \cdot \left( V^*(c_{\text{targ}}) - V^{\pi_{\theta_t}}(c_{\text{targ}}) \right) \cdot \langle \psi(c), \psi(c_{\text{targ}}) \rangle,$$

*where $V^*(c) = \max_\pi V^\pi(c)$, and $\eta_t$ is the learning of the* REINFORCE *agent.*

*Proof.* For the REINFORCE agent, the expected improvement in the training objective can be approximated as follows:

$$I_t(c) = \mathbb{E}_{\xi_t|c_t}[\Delta_t(\theta_{t+1}|\theta_t, c_t, \xi_t)] \approx \eta_t \cdot \langle g_t(c), g_t(c_{\text{targ}}) \rangle.$$

For the contextual bandit setting described above, we simplify the gradient $g_t(c)$ as follows:

$$
\begin{aligned}
g_t(c) &= [\nabla_\theta V^{\pi_\theta}(c)]_{\theta=\theta_t} \\
&= \mathbb{E}_{a \sim \pi_{\theta_t}(\cdot|s,c)} \left[ R_c(s,a) \cdot [\nabla_\theta \log \pi_\theta(a|s,c)]_{\theta=\theta_t} \right] \\
&= \pi_{\theta_t}(a_c^{\text{opt}}|s,c) \cdot R_c(s, a_c^{\text{opt}}) \cdot \left[ \nabla_\theta \log \pi_\theta(a_c^{\text{opt}}|s,c) \right]_{\theta=\theta_t} \\
&= \pi_{\theta_t}(a_c^{\text{opt}}|s,c) \cdot R_c(s, a_c^{\text{opt}}) \cdot \left( \phi(s,c,a_c^{\text{opt}}) - \mathbb{E}_{a' \sim \pi_{\theta_t}(\cdot|s,c)}[\phi(s,c,a')] \right) \\
&= \pi_{\theta_t}(a_c^{\text{opt}}|s,c) \cdot R_c(s, a_c^{\text{opt}}) \cdot (1 - \pi_{\theta_t}(a_c^{\text{opt}}|s,c)) \cdot (\phi(s,c,a_c^{\text{opt}}) - \phi(s,c,a_c^{\text{non}}))
\end{aligned}
$$

$$= \pi_{\theta_t}(a_c^{\text{opt}}|s,c) \cdot R_c(s, a_c^{\text{opt}}) \cdot (1 - \pi_{\theta_t}(a_c^{\text{opt}}|s,c)) \cdot \psi(c)$$

$$= \frac{\pi_{\theta_t}(a_c^{\text{opt}}|s,c) \cdot R_c(s, a_c^{\text{opt}})}{R_c(s, a_c^{\text{opt}})} \cdot (R_c(s, a_c^{\text{opt}}) - \pi_{\theta_t}(a_c^{\text{opt}}|s,c) \cdot R_c(s, a_c^{\text{opt}})) \cdot \psi(c)$$

$$= \frac{V^{\pi_{\theta_t}}(c)}{V^*(c)} \cdot \left(V^*(c) - V^{\pi_{\theta_t}}(c)\right) \cdot \psi(c),$$

where we used the facts that $V^{\pi_{\theta_t}}(c) = \pi_{\theta_t}(a_c^{\text{opt}}|s,c) \cdot R_c(s, a_c^{\text{opt}})$ and $V^*(c) = R_c(s, a_c^{\text{opt}})$. Based on the above simplification of $g_t(c)$, we have:

$$I_t(c)$$
$$\approx \eta_t \cdot \langle g_t(c), g_t(c_{\text{targ}}) \rangle$$
$$= \eta_t \cdot \frac{V^{\pi_{\theta_t}}(c)}{V^*(c)} \cdot \left(V^*(c) - V^{\pi_{\theta_t}}(c)\right) \cdot \frac{V^{\pi_{\theta_t}}(c_{\text{targ}})}{V^*(c_{\text{targ}})} \cdot \left(V^*(c_{\text{targ}}) - V^{\pi_{\theta_t}}(c_{\text{targ}})\right) \cdot \langle \psi(c), \psi(c_{\text{targ}}) \rangle.$$

$\square$

## C.2 TREE-STRUCTURED MDP SETTING

Let $\mathcal{M}$ denote a collection of contextual MDPs sharing a deterministic tree structure. In this structure, the state space $\mathcal{S}$ encompasses all nodes in the tree, the action space $\mathcal{A} = \{a_1, \ldots, a_N\}$ is defined such that taking an action from any non-leaf node deterministically transitions to a node in the subsequent level, while taking any action from a leaf node leads to a terminal state. The initial state is consistently set as the top node. Each MDP $\mathcal{M}_c \in \mathcal{M}$ is associated with a distinct leaf node $L_c$ serving as the goal state. Notably, we operate within a sparse reward setting, wherein taking any action from a goal state yields a reward of 1, while all other scenarios result in a reward of 0. Then, for the REINFORCE agent under the above setting, the following proposition provides an upper bound on the quantity $\|g_t(c)\|_1$:

**Proposition 1.** *For the* REINFORCE *agent with softmax policy parameterization under the contextual MDP setting described above, we have:*

$$\|g_t(c)\|_1 \leq 2 \cdot \phi_{\max} \cdot U^{\pi_{\theta_t}}(c),$$

*where* $\phi_{\max} = \max_{(s,c,a) \in \mathcal{S} \times \mathcal{C} \times \mathcal{A}} \|\phi(s,c,a)\|_1$, *and*

$$\frac{V^{\pi_{\theta_t}}(c)}{V^*(c)} \cdot (V^*(c) - V^{\pi_{\theta_t}}(c)) \leq U^{\pi_{\theta_t}}(c) \leq (V^*(c) - V^{\pi_{\theta_t}}(c)).$$

*Proof.* For the REINFORCE agent under the contextual MDP setting, we write $g_t(c)$ as follows:

$$g_t(c) = [\nabla_\theta V^{\pi_\theta}(c)]_{\theta=\theta_t} = \mathbb{E}_{\xi \sim \mathbf{Pr}_c^{\pi_{\theta_t}}} \left[ R_c(\xi) \cdot \sum_\tau \left[ \nabla_\theta \log \pi_\theta(a^{(\tau)}|s^{(\tau)}, c) \right]_{\theta=\theta_t} \right],$$

where $\xi = \left\{ (s^{(\tau)}, a^{(\tau)}) \right\}_\tau$, $\mathbf{Pr}_c^{\pi_{\theta_t}}(\xi) = \prod_\tau \pi_{\theta_t}(a^{(\tau)}|s^{(\tau)}, c)$, and $R_c(\xi) = \sum_\tau \gamma^\tau \cdot R_c(s^{(\tau)}, a^{(\tau)})$. Let $\xi_{\text{opt},c} = \left\{ (s_{\text{opt},c}^{(\tau)}, a_{\text{opt},c}^{(\tau)}) \right\}_\tau$ be the unique optimal trajectory for the MDP $\mathcal{M}_c$. Then, using the softmax policy parameterization, we can write $g_t(c)$ as follows:

$$g_t(c) = \mathbb{E}_{\xi \sim \mathbf{Pr}_c^{\pi_{\theta_t}}} \left[ \mathbf{1}[\xi = \xi_{\text{opt},c}] \cdot R_c(\xi) \cdot \sum_\tau \left[ \nabla_\theta \log \pi_\theta(a^{(\tau)}|s^{(\tau)}, c) \right]_{\theta=\theta_t} \right]$$

$$= \mathbf{Pr}_c^{\pi_{\theta_t}}(\xi_{\text{opt},c}) \cdot R_c(\xi_{\text{opt},c}) \cdot \sum_\tau \left[ \nabla_\theta \log \pi_\theta(a_{\text{opt},c}^{(\tau)}|s_{\text{opt},c}^{(\tau)}, c) \right]_{\theta=\theta_t}$$

$$= R_c(\xi_{\text{opt},c}) \cdot \mathbf{Pr}_c^{\pi_{\theta_t}}(\xi_{\text{opt},c}) \cdot \sum_\tau \phi(s_{\text{opt},c}^{(\tau)}, c, a_{\text{opt},c}^{(\tau)}) - \mathbb{E}_{a \sim \pi_{\theta_t}(\cdot|s_{\text{opt},c}^{(\tau)}, c)} \left[ \phi(s_{\text{opt},c}^{(\tau)}, c, a) \right].$$

Then, we bound the $\ell_1$-norm of $g_t(c)$ as follows:

$$\|g_t(c)\|_1 \leq R_c(\xi_{\text{opt},c}) \cdot \mathbf{Pr}_c^{\pi_{\theta_t}}(\xi_{\text{opt},c}) \cdot \sum_\tau \left\| \phi(s_{\text{opt},c}^{(\tau)}, c, a_{\text{opt},c}^{(\tau)}) - \mathbb{E}_{a \sim \pi_{\theta_t}(\cdot|s_{\text{opt},c}^{(\tau)}, c)} \left[ \phi(s_{\text{opt},c}^{(\tau)}, c, a) \right] \right\|_1$$

$$\leq R_c(\xi_{\text{opt},c}) \cdot \mathbf{Pr}_c^{\pi_{\theta_t}}(\xi_{\text{opt},c}) \cdot \sum_{\tau} \left\{ (1 - \pi_{\theta_t}(a_{\text{opt},c}^{(\tau)}|s_{\text{opt},c}^{(\tau)}, c)) \cdot \left\| \phi(s_{\text{opt},c}^{(\tau)}, c, a_{\text{opt},c}^{(\tau)}) \right\|_1 \right.$$

$$\left. + \sum_{a \neq a_{\text{opt},c}^{(\tau)}} \pi_{\theta_t}(a|s_{\text{opt},c}^{(\tau)}, c) \cdot \left\| \phi(s_{\text{opt},c}^{(\tau)}, c, a) \right\|_1 \right\}$$

$$\leq R_c(\xi_{\text{opt},c}) \cdot \mathbf{Pr}_c^{\pi_{\theta_t}}(\xi_{\text{opt},c}) \cdot \sum_{\tau} \left\{ (1 - \pi_{\theta_t}(a_{\text{opt},c}^{(\tau)}|s_{\text{opt},c}^{(\tau)}, c)) + \sum_{a \neq a_{\text{opt},c}^{(\tau)}} \pi_{\theta_t}(a|s_{\text{opt},c}^{(\tau)}, c) \right\} \cdot \phi_{\max}$$

$$= 2 \cdot \phi_{\max} \cdot R_c(\xi_{\text{opt},c}) \cdot \mathbf{Pr}_c^{\pi_{\theta_t}}(\xi_{\text{opt},c}) \cdot \sum_{\tau} (1 - \pi_{\theta_t}(a_{\text{opt},c}^{(\tau)}|s_{\text{opt},c}^{(\tau)}, c)).$$

We lower and upper bound the term $\mathbf{Pr}_c^{\pi_{\theta_t}}(\xi_{\text{opt},c}) \cdot \sum_{\tau} (1 - \pi_{\theta_t}(a_{\text{opt},c}^{(\tau)}|s_{\text{opt},c}^{(\tau)}, c))$ as follows:

$$\mathbf{Pr}_c^{\pi_{\theta_t}}(\xi_{\text{opt},c}) \cdot (1 - \mathbf{Pr}_c^{\pi_{\theta_t}}(\xi_{\text{opt},c}))$$

$$= \mathbf{Pr}_c^{\pi_{\theta_t}}(\xi_{\text{opt},c}) \cdot \sum_{\tau} \left\{ \prod_{\tau'=0}^{\tau-1} \pi_{\theta_t}(a_{\text{opt},c}^{(\tau')}|s_{\text{opt},c}^{(\tau')}, c) \right\} \cdot (1 - \pi_{\theta_t}(a_{\text{opt},c}^{(\tau)}|s_{\text{opt},c}^{(\tau)}, c))$$

$$\leq \mathbf{Pr}_c^{\pi_{\theta_t}}(\xi_{\text{opt},c}) \cdot \sum_{\tau} (1 - \pi_{\theta_t}(a_{\text{opt},c}^{(\tau)}|s_{\text{opt},c}^{(\tau)}, c)),$$

and

$$(1 - \mathbf{Pr}_c^{\pi_{\theta_t}}(\xi_{\text{opt},c})) = \sum_{\tau} \left\{ \prod_{\tau'=0}^{\tau-1} \pi_{\theta_t}(a_{\text{opt},c}^{(\tau')}|s_{\text{opt},c}^{(\tau')}, c) \right\} \cdot (1 - \pi_{\theta_t}(a_{\text{opt},c}^{(\tau)}|s_{\text{opt},c}^{(\tau)}, c))$$

$$\geq \sum_{\tau} \left\{ \prod_{\tau} \pi_{\theta_t}(a_{\text{opt},c}^{(\tau')}|s_{\text{opt},c}^{(\tau')}, c) \right\} \cdot (1 - \pi_{\theta_t}(a_{\text{opt},c}^{(\tau)}|s_{\text{opt},c}^{(\tau)}, c))$$

$$= \mathbf{Pr}_c^{\pi_{\theta_t}}(\xi_{\text{opt},c}) \sum_{\tau} (1 - \pi_{\theta_t}(a_{\text{opt},c}^{(\tau)}|s_{\text{opt},c}^{(\tau)}, c)),$$

respectively. Further, by using the fact that $V^{\pi_{\theta_t}}(c) = R_c(\xi_{\text{opt},c}) \cdot \mathbf{Pr}_c^{\pi_{\theta_t}}(\xi_{\text{opt},c})$ and $V^*(c) = R_c(\xi_{\text{opt},c})$, we have:

$$R_c(\xi_{\text{opt},c}) \cdot \mathbf{Pr}_c^{\pi_{\theta_t}}(\xi_{\text{opt},c}) \cdot (1 - \mathbf{Pr}_c^{\pi_{\theta_t}}(\xi_{\text{opt},c})) = \frac{V^{\pi_{\theta_t}}(c)}{V^*(c)} \cdot (V^*(c) - V^{\pi_{\theta_t}}(c)),$$

and

$$R_c(\xi_{\text{opt},c}) \cdot (1 - \mathbf{Pr}_c^{\pi_{\theta_t}}(\xi_{\text{opt},c})) = (V^*(c) - V^{\pi_{\theta_t}}(c)).$$

Finally, by combining these results, we have the following:

$$\|g_t(c)\|_1 \leq 2 \cdot \phi_{\max} \cdot U^{\pi_{\theta_t}}(c),$$

where

$$\frac{V^{\pi_{\theta_t}}(c)}{V^*(c)} \cdot (V^*(c) - V^{\pi_{\theta_t}}(c)) \leq U^{\pi_{\theta_t}}(c) \leq (V^*(c) - V^{\pi_{\theta_t}}(c)).$$

$\square$

# D  EXPERIMENTAL EVALUATION – ADDITIONAL DETAILS

## D.1  ENVIRONMENTS

**Sparse Goal Reaching (SGR (Klink et al., 2022)).** In this environment, the state consists of the agent's position, denoted as $s = [x\, y]$. The action corresponds to the agent's displacement in a 2D space, represented as $a = [d_x\, d_y]$. The contextual variable $c = [c_1\, c_2\, c_3] \in \mathcal{C} \subseteq \mathbb{R}^3$ comprises the following elements: *C-GoalPositionX*, *C-GoalPositionY*, and *C-Tolerance*. The bounds for each contextual variable are $[-9, 9]$ for *C-GoalPositionX*, $[-9, 9]$ for *C-GoalPositionY*, and $[0.05, 18]$ for *C-Tolerance*. The reward in this environment is sparse, meaning the agent receives a reward of

1 only when it reaches the goal. An episode is considered successful if the distance between the agent's position and the goal is below the tolerance, i.e., $\|s - [c_1, c_2]\|_2 \leq c_3$. If the agent exceeds the limit of 200 steps per episode before reaching the goal, the episode terminates with a reward of 0. The presence of walls in the environment creates situations where certain combinations of contexts $[c_1 \, c_2 \, c_3]$ are unsolvable by the agent, as it is unable to get close enough to the goal to satisfy the tolerance condition. This suggests that a successful curriculum technique should also be able to identify the feasible subspace of contexts to accelerate the training process. The target context distribution consists of tasks that are uniformly distributed w.r.t. the contexts $c_1$ (*C-GoalPositionX*) and $c_2$ (*C-GoalPositionY*). However, it is concentrated in the minimal *C-Tolerance*, where $c_3$ is set to 0.05. In other words, the target distribution comprises only high-precision tasks.

In this environment, we employ Proximal Policy Optimization (PPO) with 5120 steps per policy update and a batch size of 256. The MLP policy consists of a shared layer with 64 units, followed by a second separate layer with 32 units for the policy network, and an additional 32 units for the value network. All the remaining parameters of PPO adopt the default settings of Stable Baselines 3 (Schulman et al., 2017; Raffin et al., 2021). Furthermore, all the hyperparameters remain consistent across all the curriculum strategies evaluated.

**Point Mass Sparse (POINTMASS-S (Klink et al., 2020b)).** In this environment, the state consists of the position and velocity of the point-mass, denoted as $s = [x \, \dot{x} \, y \, \dot{y}]$. The action corresponds to the force applied to the point-mass in a 2D space, represented as $a = [F_x \, F_y]$. The contextual variable $c = [c_1 \, c_2 \, c_3] \in \mathcal{C} \subseteq \mathbb{R}^3$ comprises the following elements: *C-GatePosition*, *C-GateWidth*, and *C-Friction*. The bounds for each contextual variable are $[-4, 4]$ for *C-GatePosition*, $[0.5, 8]$ for *C-GateWidth*, and $[0, 4]$ for *C-Friction*. At the beginning of each episode, the agent's initial state is set to $s_0 = [0 \, 0 \, 3 \, 0]$, and the objective is to approach the goal located at position $g = [x \, y] = [0 \, -3]$ with sufficient proximity. If the agent collides with a wall or if the episode exceeds 100 steps, the episode is terminated, and the agent receives a reward of 0. On the other hand, if the agent reaches the goal within a predefined threshold, specifically when $\|g - [x \, y]\|_2 < 0.30$, the episode is considered successful, and the agent receives a reward of 1. The target distribution $\mu$ is represented by a bimodal Gaussian distribution, with the means of the contexts $[\text{\textit{C-GatePosition}} \, \text{\textit{C-GateWidth}}]$ set as $[-3.9 \, 0.5]$ and $[3.9 \, 0.5]$ for the two modes, respectively. This choice of target distribution presents a challenging scenario, as it includes contexts where the gate's position is in proximity to the edges of the environment and the gate's width is relatively small.

In this environment, we employ Proximal Policy Optimization (PPO) with 5120 steps per policy update. The batch size used for each update is set to 128, and an entropy coefficient of 0.01 is applied. The MLP policy consists of a shared layer with 64 units, followed by a second separate layer with 64 units for the policy network, and an additional 64 units for the value network. All the remaining parameters of PPO adopt the default settings of Stable Baselines 3 (Schulman et al., 2017; Raffin et al., 2021). Furthermore, all the hyperparameters remain consistent across all the curriculum strategies evaluated.

**Bipedal Walker Stump Tracks BIPEDALWALKER (Romac et al., 2021)).** The experiment with BIPEDALWALKER is based on the stump tracks environment with a classic bipedal walker embodiment, as can be found in TeachMyAgent benchmark (Romac et al., 2021). The experimental setting is considered a mostly trivial task space, and the curriculum technique (teacher component) has no prior knowledge. Namely, in this setting, no reward mastery range, no prior knowledge concerning the task space, i.e., regions containing trivial tasks, and no subspace of test tasks are given. The learned policy that controls the walker agent with motor torque is trained with the Soft Actor Critic (SAC) algorithm for 20 million steps.

### D.2 CURRICULUM STRATEGIES EVALUATED

**Variants of our curriculum strategy.** Below, we report the hyperparameters and implementation details of the variants of our curriculum strategies used in the experiments (for each environment):

1. For both PROXCORL and PROXCORL-UN:

   (a) $\beta$ parameter controls the softmax selection's stochasticity: we set $\beta = 50$ for the SGR and the BIPEDALWALKER environment, and $\beta = 70$ for POINTMASS-S:2G and POINTMASS-S:1T.

    (b) $V_{\max}$ normalization parameter: we set $V_{\max} = 1$ for all environments. Since BIPEDAL-WALKER is a dense reward environment, the reward is scaled with the upper and lower bound rewards as provided by Romac et al. (2021).

    (c) $N_{\mathrm{pos}}$ parameter that controls the frequency at which $V^t(\cdot)$ is updated: we set $N_{\mathrm{pos}} = 5120$ for SGR, POINTMASS-S:2G and POINTMASS-S:1T environments, which is equivalent to the PPO update frequency. For BIPEDALWALKER, the frequency of updates is after each episode.

**State-of-the-art baselines.** Below, we report the hyperparameters and implementation details of the state-of-the-art curriculum strategies used in the experiments (for each environment):

1. For SPDL (Klink et al., 2020b):

    (a) $V_{\mathrm{LB}}$ performance threshold: we set $V_{\mathrm{LB}} = 0.5$ for SGR, POINTMASS-S:2G, and POINTMASS-S:1T environment.

    (b) $N_{\mathrm{pos}}$ parameter that controls the frequency of performing the optimization step to update the distribution for selecting tasks: we set $N_{\mathrm{pos}} = 5120$ for SGR, POINTMASS-S:2G, and POINTMASS-S:1T environment.

    (c) For BIPEDALWALKER, we perform the experiments provided in Romac et al. (2021) for the Self-Paced teacher, which is equivalent to SPDL technique.

2. For CURROT (Klink et al., 2022):

    (a) $V_{\mathrm{LB}}$ performance threshold: we set $V_{\mathrm{LB}} = 0.4$ for SGR, $V_{\mathrm{LB}} = 0.6$ for POINTMASS-S:2G and POINTMASS-S:1T, and $V_{\mathrm{LB}} = 180$ for BIPEDALWALKER.

    (b) $\epsilon$ distance threshold between subsequent distributions: we set $\epsilon = 1.5$ for SGR, $\epsilon = 0.05$ for POINTMASS-S:2G and POINTMASS-S:1T, and $\epsilon = 0.5$ for BIPEDAL-WALKER.

    (c) We choose the best-performing pair $(V_{\mathrm{LB}}, \epsilon)$ for each environment from the set $\{0.4, 0.5, 0.6\} \times \{0.05, 0.5, 1.0, 1.5, 2.0\}$. For BIPEDALWALKER, we use the hyperparameters provided in Klink et al. (2022).

    (d) $N_{\mathrm{pos}}$ parameter that controls the frequency of performing the optimization step to update the distribution for selecting tasks: we set $N_{\mathrm{pos}} = 5120$ for all environments.

    (e) The implementation in this paper incorporates all the original components of the strategy, including the update of the success buffer, the computation of the updated context distribution, and the utilization of a Gaussian mixture model to search for contexts that meet the performance threshold.

    (f) At the beginning of the training process, the initial contexts are uniformly sampled from the context space $\mathcal{C}$, following the same approach utilized in all other techniques.

3. For PLR (Jiang et al., 2021b):

    (a) $\rho$ staleness coefficient: we set $\rho = 0.5$ for the SGR, POINTMASS-S:2G and POINTMASS-S:1T environment.

    (b) $\beta_{\mathrm{PLR}}$ temperature parameter for score prioritization: we set $\beta_{\mathrm{PLR}} = 0.1$ for all the environments.

    (c) $N_{\mathrm{pos}}$ parameter that controls the frequency at which $V^t(\cdot)$ is updated: we set $N_{\mathrm{pos}} = 5120$ for all the environments.

    (d) The technique has been adapted to operate with a pool of tasks. By employing a binomial decision parameter $d$, a new, unseen task is randomly selected from the task pool and added to the set of previously seen tasks. The seen tasks are prioritized and chosen based on their learning potential upon revisiting, aligning with the approach utilized in the original strategy. As more unseen tasks are sampled from the pool, the binomial decision parameter $d$ is gradually annealed until all tasks are seen. Once this occurs, only the seen tasks are sampled from the replay distribution, taking into account their learning potential.

4. For GRADIENT (Huang et al., 2022):

    (a) Number of stages $N_{\mathrm{stage}}$: we set 5 stages for SGR, and 10 stages for POINTMASS-S:2G, POINTMASS-S:1T, and BIPEDALWALKER. We selected the number of stages based on the original paper experiments and a value search in the set $\{3, 5, 7, 10\}$.

(b) Maximum number of steps per stage: we select $100000$ steps as the maximum number of training steps before switching to the next stage.

(c) $\Delta\alpha_{\text{GRADIENT}}$ per stage: we set $\Delta\alpha = 0.2$ for SGR, and $\Delta\alpha = 0.1$ for BIPEDAL-WALKER. For POINTMASS-S:2G and POINTMASS-S:1T, we choose the next $\alpha$ based on $\alpha(i) = \frac{1}{N_{\text{stage}} - i}$. These selections are based on the experimental section and the provided implementation from Huang et al. (2022).

(d) Reward threshold per stage is set to $0.8$ for all the experiments. If the policy achieves this threshold, it switches to the next stage before reaching the maximum number of steps.

Our curriculum strategy only requires forward-pass operation on the critic-model to obtain value estimates for a subset of tasks $c$ and $c_{\text{targ}}$ in $\mathcal{C}$, followed by an $\arg\max$ operation over this subset. We note that the computational overhead of our curriculum strategy is minimal compared to the baselines. In particular, SPDL and CURROT require the same forward-pass operations and perform an additional optimization step to obtain the next task distribution. CURROT relies on solving an Optimal Transport problem requiring a high computational cost. Even when reducing the Optimal Transport problem to a linear assignment problem, as done in practice, the complexity is $O(n^3)$. As for PLR, it has an additional computational overhead for scoring the sampled tasks.

