# OpenReview forum: "Proximal Curriculum with Task Correlations for Deep Reinforcement Learning"
_ICLR.cc/2024/Conference — Submitted to ICLR 2024_

### Official Review · Reviewer_MJvx · 2023-10-21

**Soundness:** 2 fair
**Presentation:** 3 good
**Contribution:** 2 fair
**Rating:** 5
**Confidence:** 3

**Summary:**

This work presents a curriculum learning method for multi-task reinforcement learning agents to solve a target distribution of tasks. It balances between selecting tasks of moderate difficulty for the learning agent to solve and proposing tasks that are similar to the target distribution. The metric for sampling tasks is derived in a specific learning scenario and is then applied to other more general settings. In implementation, the method involves sampling a discrete pool of tasks from the uniform and target distribution of tasks and prioritizing the learning of the tasks proportional to the proposed metric. The experiments are conducted in both binary sparse reward and dense reward environments to solve various target distributions of tasks.

**Strengths:**

1. The method can deal with curriculum learning towards various target distributions of tasks, different to other works that consider a uniform target distribution.
2. The paper is generally well-written and easy to follow.

**Weaknesses:**

1. It is not unconvincing to directly apply the curriculum strategy derived from the simple scenario in Sec 3.1 to the general case in Sec 3.2, since the simple scenario is with specific action and reward settings.
2. The metric for task similarity is a bit specific to the tasks. Currently, the metric is defined based on the L2 distance of context parameters. However, there are many cases where such a metric is not positively correlated with the intuitive similarity between tasks. For example, consider a table-top manipulation environment with a drawer and a cube on the table. Suppose the initial state is with the drawer closed and the cube on the table outside the drawer. Consider a desired state A with the cube on top of the closed drawer and a desired state B with the cube inside the closed drawer. Suppose the context parameters are defined as the open width of the drawer and the pose of the cube. We can see that the L2 distance between A and B is small, but the complexity of strategies to reach A and B is different: reaching state A only requires one pick-and-place while reaching state B requires opening the drawer, placing the cube, and then closing the drawer. It is inappropriate for the curriculum to treat these tasks as similar, and I think a better metric for task similarity is required.
3. Fig. 2(b) shows that PROXCORL performs similarly to PROXCORL-UN in PointMass-s:2G environment. Since the target task distribution is non-uniform in this environment, the result weakens the contribution of the proposed curriculum that takes task correlation into consideration.

**Questions:**

Please refer to the "weaknesses" part.

---

> ### Author Response · Authors · 2023-11-19
> **Response to Reviewer MJvx**
>
> Thank you for carefully reviewing our paper! We greatly appreciate your feedback. Please see below our responses to your comments.
>
> -----
>
> **1. Extending the presented curriculum analysis for the contextual bandit setting to a general RL setting.**
>
> Our investigation into developing a curriculum strategy for the multi-task RL setting (outlined in Section 2) begins by observing that the following three essential factors must be taken into account when selecting a source-target task pair for the student component: (i) Learning potential in the source task, (ii) Transfer potential of the source task, i.e., similarity with the target task, and (iii) Potential performance gain in the target task.
>
> The challenge lies in integrating these quantities to devise an effective curriculum strategy. In pursuit of this goal, we opt to analyze a contextual bandit setting. Intriguingly, our analysis leads us to an intuitive curriculum strategy (Eq. (1)), involving the geometric mean of the three aforementioned factors. We extend this form to general learning settings with natural modifications. Furthermore, our empirical investigation demonstrates the effectiveness of our curriculum strategy.
>
> In the revised version of this paper, we present a high-level approximation for the expected improvement in the training objective in a general RL setting (Section 3.1). Additionally, in the Appendix C.2, we conduct an extra analysis of a tree-structured contextual MDP setting. This analysis underscores the significance of the ZPD-related terms in our proposed curriculum strategy.
>
> While it remains intriguing to delve into more complex learning settings and explore the potential for uncovering richer curriculum strategies, we acknowledge the general absence of such analyses in the existing literature.
>
> -----
>
> **2. The metric for task similarity is a bit specific to the tasks. Currently, the metric is defined based on the L2 distance of context parameters.**
>
> We thank the reviewer for pointing this out; it is indeed an interesting observation. In our work, we design our curriculum strategy to be agnostic to the task similarity metric, allowing for the application of any such metric. Nevertheless, for the purpose of the empirical evaluation, we opted to employ a standard similarity metric consistently across all domains. Exploring more generalized similarity metrics spanning various domains or even learning the similarity metric itself represents an interesting avenue for future research.
>
> -----
>
> **3. Clarification regarding the narrow performance gap between ProxCoRL and ProxCoRL-Un in the PointMass-s:2G environment with a non-uniform target distribution.**
>
> Based on the experimental results presented in the paper, we hypothesize that, for the PointMass-s:2G environment, the first two quantities (1) and (2) in Eq. (1), aligned with the ZPD principle, are enough to shift the training task distribution toward harder tasks. This observation aligns with similar trends identified in [1]. Through this shift towards harder tasks, the training task distribution ultimately converges to the target task distribution, even without explicit knowledge of the latter.
>
> Conversely, in the PointMass-s:1T environment, where the target distribution is non-uniform, we observe that the performance gap between ProxCoRL and ProxCoRL-Un increases. Specifically, ProxCoRL converges more rapidly to the target distribution compared to ProxCoRL-Un. This indicates that, in the single target task setting, the incorporation of terms (3) and (4) intensifies the preference for tasks from the target distribution. This increased preference contributes to a more substantial distribution shift towards the target distribution, consequently improving the agent's speed in solving the target task.
>
> -----
>
> [1] Georgios Tzannetos, B´arbara Gomes Ribeiro, Parameswaran Kamalaruban, and Adish Singla. Proximal Curriculum for Reinforcement Learning Agents. In TMLR, 2023.
>
> -----
>
> We hope that our responses can address your concerns and are helpful in improving your rating. If you have any other comments or feedback, please let us know! We are looking forward to hearing back from you! Thank you again for the review.

---

> > ### Comment · Reviewer_MJvx · 2023-11-22
> >
> > Dear authors,
> >
> > Thank you for the rebuttal. My concern in weakness 2 is addressed.
> >
> > For weakness 3, I still feel confused about why terms (1) and (2) in Eq. 1 can shift the distribution to a seemingly non-trivial target distribution (bimodal Gaussian) in PointMass-s:2G while not to a delta distribution in PointMass-s:1T. Could you provide more evidence?
> >
> > For weakness 1, I appreciate the added analysis, but the theoretical derivation to general RL settings still requires more studies.
> >
> > Overall, I would like to keep my score unchanged.

---

### Official Review · Reviewer_LEfH · 2023-10-26

**Soundness:** 2 fair
**Presentation:** 3 good
**Contribution:** 2 fair
**Rating:** 5
**Confidence:** 3

**Summary:**

This paper considers the curriculum learning setting in reinforcement learning problems. It focuses on the selection strategy of tasks from the target task distribution. The goal is to maximize the expected return of the learned policy on target task distribution. This work approximates this objective in a simplified learning setting (a single state and two actions in contextual MDP) with a single target task. And the approximation yields a curriculum strategy. According to the theoretical derivation, the authors propose the task selection strategy in the general setting of contextual MDP with arbitrary target task distributions.

The experiments are conducted on the tasks of sparse goal-reaching and bipedal walker stump tracks. The proposed method almost outperforms all baselines, including SOTA algorithms for curriculum learning.

**Strengths:**

The proposed method is well-motivated with a theoretical foundation.

This paper is generally well-written and the proposed approach is clearly presented.

**Weaknesses:**

The theoretical contribution is not strong enough, since the theorem is derived from a super simplified setting. And the adaptation to more general setting is not directly supported by the theorem.

The proposed algorithm is not obviously better than the baselines.

The experiments are only conducted on relatively simple tasks with state-based policy. It will be much more impressive if the proposed method can handle vision-baed RL policy.

**Questions:**

As for the theoretical derivation in Section 3, is it possible to make the theoretical contribution stronger? Can we extend Theorem 1 by relaxing its assumption about the MDP or the target task distribution? The theoretical derivation looks a bit trivial when only considering the contextual MDP with singleton state space and a single target task.

As for the changes from Equation 1 to Equation 2, why we can replace V*() with V_{max}? Is there any mathematical derivation behind it?

---

> ### Author Response · Authors · 2023-11-19
> **Response to Reviewer LEfH**
>
> Thank you for carefully reviewing our paper! We greatly appreciate your feedback. Please see below our responses to your comments.
>
> -----
>
> **1. Extending the presented curriculum analysis for the contextual bandit setting to a general RL setting.**
>
> Our investigation into developing a curriculum strategy for the multi-task RL setting (outlined in Section 2) begins by observing that the following three essential factors must be taken into account when selecting a source-target task pair for the student component: (i) Learning potential in the source task, (ii) Transfer potential of the source task, i.e., similarity with the target task, and (iii) Potential performance gain in the target task.
>
> The challenge lies in integrating these quantities to devise an effective curriculum strategy. In pursuit of this goal, we opt to analyze a contextual bandit setting. Intriguingly, our analysis leads us to an intuitive curriculum strategy (Eq. (1)), involving the geometric mean of the three aforementioned factors. We extend this form to general learning settings with natural modifications. Furthermore, our empirical investigation demonstrates the effectiveness of our curriculum strategy.
>
> In the revised version of this paper, we present a high-level approximation for the expected improvement in the training objective in a general RL setting (Section 3.1). Additionally, in the Appendix C.2, we conduct an extra analysis of a tree-structured contextual MDP setting. This analysis underscores the significance of the ZPD-related terms in our proposed curriculum strategy.
>
> -----
>
> **2. The proposed algorithm is not obviously better than the baselines.**
>
> In the empirical evaluation we made, the proposed curriculum is better or at least comparable with the baselines across all the environments.
>
> In the SGR environment, Fig (2a), ProxCoRL outperforms the other techniques.
>
> In the PointMass-s:2G environment, Fig (2b), we can observe a strong performance of ProxCoRL-Un, equivalent to the proposed technique. We hypothesize that, for the PointMass-s:2G environment, the first two quantities (1) and (2) in Eq. (1), aligned with the ZPD principle, are enough to shift the training task distribution toward harder tasks. This observation aligns with similar trends identified in [1]. Through this shift towards harder tasks, the training task distribution ultimately converges to the target task distribution, even without explicit knowledge of the latter.
>
> In the PointMass-s:1T environment, Fig (2c), where the target distribution is non-uniform, we observe that the performance gap between ProxCoRL and ProxCoRL-Un increases. Specifically, ProxCoRL converges more rapidly to the target distribution compared to ProxCoRL-Un. This indicates that, in the single target task setting, the incorporation of terms (3) and (4) intensifies the preference for tasks from the target distribution. This increased preference contributes to a more substantial distribution shift towards the target distribution, consequently improving the agent's speed in solving the target task. CURROT and GRADIENT, both of which are curriculum strategies capable of converging to a single target, demonstrate effectiveness in this environment. Notably, ProxCoRL exhibits a faster convergence rate.
>
> In the BipedalWalker environment, Fig (2d), the uniform nature of the target distribution suggests that ProxCoRL-Un should perform well in this setting. Interestingly, despite the uniform target distribution, ProxCoRL performs comparably to ProxCoRL-Un.
>
> -----
>
> **3. The experiments are only conducted on relatively simple tasks with state-based policy.**
>
> The experiments are carried out in environments commonly employed in state-of-the-art curriculum techniques ([2], [3]). For example, the BipedalWalker environment features a high-dimensional state space in $R^{24}$ and incorporates LIDAR measurements.
>
> -----
>
> **4. Rationale behind replacing $V^\*()$ with $V_{\max}$ in the practical curriculum strategy.**
>
> In the general practical setting, we simplify by assuming that the optimal value $V^\*()$ of the environment can be replaced with the maximum achievable value $V_{\max}$ within this environment. In goal-based sparse reward reinforcement learning environments, this value is commonly set to $1$. In domains where obtaining a prior estimate of $V_{\max}$ proves challenging, a practical approach involves dynamically estimating it during training. This is achieved by continuously monitoring and updating the running maximum of the observed returns.
>
> -----
>
> [1] Tzannetos et al. Proximal Curriculum for Reinforcement Learning Agents. In TMLR, 2023.
>
> [2] Klink et al. Curriculum Reinforcement Learning via Constrained Optimal Transport. In ICML, 2022.
>
> [3] Romac et al. Teachmyagent: a Benchmark for Automatic Curriculum Learning in Deep RL. In ICML, 2021.
>
> -----
>
> We hope that our responses can address your concerns and are helpful in improving your rating. Thank you again for the review.

---

> ### Comment · Reviewer_LEfH · 2023-11-20
> **Thanks for the detailed response**
>
> Thank authors for the detailed response. I appreciate your efforts on the newly updated theoretical results in C.2. However, I'm still concerned that the proposed method is not significantly better than baselines and the theoretical contribution is super limited to over-simplified cases. Thus, I'd keep my score.

---

### Official Review · Reviewer_eLrA · 2023-10-31

**Soundness:** 3 good
**Presentation:** 3 good
**Contribution:** 4 excellent
**Rating:** 8
**Confidence:** 3

**Summary:**

* This paper introduces a curriculum strategy that applies the “Zone of Proximal Development” concept to accelerate the learning progress of RL agents.

**Strengths:**

* The paper is well-written.
* The method is motivated well; the proper background is introduced, and the technical details are clear.
* The paper provides good context for related work, and the appropriate baselines are used for evaluation.
* The results are convincing.

**Weaknesses:**

* Use “citep” instead of “citet” for citations where the citation is not part of the sentence. In the Section 2, there are many instances of this error (e.g. “Hallack et al., 2015…”, “Sutton et al., 1999).
* The above error occurs in following sections as well (see paragraph 3 in Section 3.1; First paragraph in Section 4).
* Minor: Figure 2 plotlines are a bit thick, making it somewhat difficult to read. I would suggest slightly decreasing the line width.

**Questions:**

* None.

---

> ### Author Response · Authors · 2023-11-19
> **Response to Reviewer eLrA**
>
> Thank you for carefully reviewing our paper! We greatly appreciate your feedback. Please see below our responses to your comments.
>
> -----
>
> **1. Use “citep” instead of “citet” for citations where the citation is not part of the sentence… Minor: Figure 2 plotlines are a bit thick, making it somewhat difficult to read. I would suggest slightly decreasing the line width.**
>
> We thank the reviewer for these comments. We have updated the PDF with the suggested changes.
>
> -----
>
> If you have any other comments or feedback, please let us know! We are looking forward to hearing back from you! Thank you again for the review.

---

### Official Review · Reviewer_ynrZ · 2023-11-07

**Soundness:** 3 good
**Presentation:** 3 good
**Contribution:** 2 fair
**Rating:** 3
**Confidence:** 4

**Summary:**

The authors suggest a curriculum approach to reinforcement learning within a contextual multi-task framework. Initially, this framework is used to formulate a teacher-student curriculum learning strategy, and the authors analyze the learning objective from the perspective of the teacher. Following this, the authors propose a strategy called ProxCoRL, which selects tasks $c_t\$ based on their analysis. The paper ultimately demonstrates that the proposed strategy empirically outperforms previous curriculum-based reinforcement learning methods and baseline algorithms across a variety of tasks.

**Strengths:**

The paper emphasizes the significance of a balanced task selection strategy, which ensures that the tasks presented to the agent are neither too difficult nor too simple. This strategy facilitates the agent's progressive learning towards the target distribution. The approach employs a task selection mechanism and is supported by a mathematical analysis within a simplified setting.

**Weaknesses:**

The paper does not present a method that is fundamentally different from those in prior work[1]. The authors attempt to extend the idea proposed in ProCuRL[1] to a general target distribution, yet the core of this extension appears to be a mere application of an existing concept. Moreover, the simplified setting used to conduct the mathematical analysis diverges significantly from a typical RL setting, given it encompasses only two possible actions and a single isolated state space. This raises questions about the scalability of such an analysis in a general reinforcement learning framework. Additionally, while the authors claim that their proposed approach eliminates the need for domain-specific hyperparameter tuning, it nonetheless requires the determination of $V_{max}$, which represents the maximum possible value.

[1] Georgios Tzannetos, B´arbara Gomes Ribeiro, Parameswaran Kamalaruban, and Adish Singla. Proximal Curriculum for Reinforcement Learning Agents. In TMLR, 2023.

**Questions:**

1. Answer the concerns in the above weakness section.

2. While the authors assert that their proposed method, ProxCoRL, is robust to the target distribution $\mu$ in contrast to ProxCuRL, the results presented in Figure 2 do not substantiate this claim. It is evident that ProxCoRL does not demonstrate effectiveness as the performance gap between ProxCoRL and ProxCuRL narrows in the case of a non-uniform task distribution (PointMass-s:2G). Could you clarify these results?

---

> ### Author Response · Authors · 2023-11-19
> **Response to Reviewer ynrZ**
>
> Thank you for carefully reviewing our paper! We greatly appreciate your feedback. Please see below our responses to your comments.
>
> -----
>
> **1. Difference between the proposed method and the one presented in prior work [1].**
>
> The setting considered in the prior work [1] differs notably. Specifically, it does not account for a target task distribution, opting instead to gauge the agent's performance relative to a uniform distribution of tasks across the context space. In our work, the curriculum is developed with consideration for a target task setting, resulting in the introduction of additional terms (3) and (4) in Eq. (1). These terms enhance the preference for selecting tasks correlated with the target task.
>
> -----
>
> **2. Extending the presented curriculum analysis for the contextual bandit setting to a general RL setting.**
>
> Our investigation into developing a curriculum strategy for the multi-task RL setting (outlined in Section 2) begins by observing that the following three essential factors must be taken into account when selecting a source-target task pair for the student component: (i) Learning potential in the source task, (ii) Transfer potential of the source task, i.e., similarity with the target task, and (iii) Potential performance gain in the target task.
>
> The challenge lies in integrating these quantities to devise an effective curriculum strategy. In pursuit of this goal, we opt to analyze a contextual bandit setting. Intriguingly, our analysis leads us to an intuitive curriculum strategy (Eq. (1)), involving the geometric mean of the three aforementioned factors. We extend this form to general learning settings with natural modifications. Furthermore, our empirical investigation demonstrates the effectiveness of our curriculum strategy.
>
> In the revised version of this paper, we present a high-level approximation for the expected improvement in the training objective in a general RL setting (Section 3.1). Additionally, in the Appendix C.2, we conduct an extra analysis of a tree-structured contextual MDP setting. This analysis underscores the significance of the ZPD-related terms in our proposed curriculum strategy.
>
> While it remains intriguing to delve into more complex learning settings and explore the potential for uncovering richer curriculum strategies, we acknowledge the general absence of such analyses in the existing literature.
>
> -----
>
> **3. The proposed method requires domain-specific hyperparameter tuning of $V_{\max}$.**
>
> $V_{\max}$ denotes the maximum attainable value in the given environment. In goal-based sparse reward reinforcement learning environments, this value is commonly set to $1$. In domains where obtaining a prior estimate of $V_{\max}$ proves challenging, a practical approach involves dynamically estimating it during training. This is achieved by continuously monitoring and updating the running maximum of the observed returns.
>
> -----
>
> **4. Clarification regarding the narrow performance gap between ProxCoRL and ProxCoRL-Un in the case of a non-uniform task distribution (PointMass-s:2G).**
>
> Based on the experimental results presented in the paper, we hypothesize that, for the PointMass-s:2G environment, the first two quantities (1) and (2) in Eq. (1), aligned with the ZPD principle, are enough to shift the training task distribution toward harder tasks. This observation aligns with similar trends identified in [1]. Through this shift towards harder tasks, the training task distribution ultimately converges to the target task distribution, even without explicit knowledge of the latter.
>
> Conversely, in the PointMass-s:1T environment, where the target distribution is non-uniform, we observe that the performance gap between ProxCoRL and ProxCoRL-Un increases. Specifically, ProxCoRL converges more rapidly to the target distribution compared to ProxCoRL-Un. This indicates that, in the single target task setting, the incorporation of terms (3) and (4) intensifies the preference for tasks from the target distribution. This increased preference contributes to a more substantial distribution shift towards the target distribution, consequently improving the agent's speed in solving the target task.
>
> -----
>
> [1] Georgios Tzannetos, B´arbara Gomes Ribeiro, Parameswaran Kamalaruban, and Adish Singla. Proximal Curriculum for Reinforcement Learning Agents. In TMLR, 2023.
>
> -----
>
> We hope that our responses can address your concerns and are helpful in improving your rating. If you have any other comments or feedback, please let us know! We are looking forward to hearing back from you! Thank you again for the review.

---

### Meta-Review · Area_Chair_Vvtd · 2023-12-11

**Metareview:**

The paper introduces ProxCoRL, a novel curriculum learning strategy for reinforcement learning (RL) agents in contextual multi-task settings. Inspired by the Zone of Proximal Development, ProxCoRL accelerates RL agent learning for a target distribution over complex tasks, leveraging task correlations. The method outperforms state-of-the-art baselines in various domains. The paper also explores a teacher-student curriculum strategy and curriculum learning in RL, demonstrating theoretical justification and empirical effectiveness. Additionally, a task-balancing method is proposed, showcasing effectiveness in solving diverse target task distributions in binary sparse and dense reward environments.

The method is well-motivated with a strong theoretical foundation and is capable of dealing with various target distributions, setting it apart from works focused on uniform distributions. The paper is commendably well-written, clearly presenting the proposed approach, providing a solid theoretical background, and effectively contextualizing related work. The results are convincing, contributing to the overall strength and credibility of the research.

On the other hand, the paper has several weaknesses. The theoretical contribution is questioned due to its derivation from a super-simplified setting, and doubts about its direct applicability to a more general context are raised. The proposed algorithm's superiority over baselines is not convincingly demonstrated, and concerns exist about its practical effectiveness. The experimental design faces criticism for the potential inaccuracy of directly applying the curriculum strategy from a simple scenario to a more general case. The metric for task similarity is considered too specific and may not align with intuitive task similarities in certain scenarios. Additionally, the absence of experiments on vision-based RL policies limits the generalizability of the proposed method.

The rejection is recommended due to the significant weaknesses mentioned above.
We encourage the authors to address these issues while preparing a new version of their paper.

**Justification For Why Not Higher Score:**

The rejection is recommended due to significant weaknesses in the paper's theoretical foundation, algorithmic performance, experimental design, and generalizability.

**Justification For Why Not Lower Score:**

N/A

---

### Decision · Program_Chairs · 2024-01-16

Reject